# Assessment of transparency indicators in space medicine

Rosa Katia Bellomo[1,2]*, Emmanuel A. Zavalis[3,4], John P. A. Ioannidis[1,4]*

**1** Meta-Research Innovation Center at Stanford (METRICS), Stanford University, Stanford, CA, United States of America, **2** Department of Public Health and Infectious Diseases, Sapienza University of Rome, Rome, Italy, **3** Department of Learning Informatics Management and Ethics, Karolinska Institutet, Stockholm, Sweden, **4** Departments of Medicine, of Epidemiology and Population Health, of Biomedical Data Science, and of Statistics, Stanford University, Stanford, CA, United States of America

\* rkatia.bellomo@gmail.com (RKB); jioannid@stanford.edu (JPAI)

## Abstract

Space medicine is a vital discipline with often time-intensive and costly projects and constrained opportunities for studying various elements such as space missions, astronauts, and simulated environments. Moreover, private interests gain increasing influence in this discipline. In scientific disciplines with these features, transparent and rigorous methods are essential. Here, we undertook an evaluation of transparency indicators in publications within the field of space medicine. A meta-epidemiological assessment of PubMed Central Open Access (PMC OA) eligible articles within the field of space medicine was performed for prevalence of code sharing, data sharing, pre-registration, conflicts of interest, and funding. Text mining was performed with the rtransparent text mining algorithms with manual validation of 200 random articles to obtain corrected estimates. Across 1215 included articles, 39 (3%) shared code, 258 (21%) shared data, 10 (1%) were registered, 110 (90%) contained a conflict-of-interest statement, and 1141 (93%) included a funding statement. After manual validation, the corrected estimates for code sharing, data sharing, and registration were 5%, 27%, and 1%, respectively. Data sharing was 32% when limited to original articles and highest in space/parabolic flights (46%). Overall, across space medicine we observed modest rates of data sharing, rare sharing of code and almost non-existent protocol registration. Enhancing transparency in space medicine research is imperative for safeguarding its scientific rigor and reproducibility.

## Introduction

Space medicine aims to support human space exploration and encompasses a wide range of research efforts. It studies the effects that physics concepts like microgravity, radiation, isolation, and pressure have on human physiology, not only during the time astronauts spend in space, but also evaluating the long-term effects of these exposure on the human body after returning on Earth [1]. Space exploration remains an open and active area of interest in science, not only for the "space race" features but also because of the practical applications that this field can provide the terrestrial world with [2]. A lot of literature has been published on

**Data Availability Statement:** The analyzed information dataset to perform all analyses in the paper is available at OSF (www.doi.org/10.17605/OSF.IO/YXU9Q). Code sharing: The underlying code for this study is available on Open Science

Framework and can be accessed via this link www.
doi.org/10.17605/OSF.IO/YXU9Q.

**Funding:** The work of John Ioannidis is supported
by an unrestricted gift from Sue and Bob O'Donnell
to Stanford. The work of Rosa Katia Bellomo is
funded within the framework of the project "ExACT
- European network staff eXchange for integrating
precision health in the health Care sysTems" (Grant
Agreement n. 823995) funded by the European
Commission under the H2020 – Marie Slodowska
Curie Action – RISE scheme. The funders had no
role in study design, data collection and analysis,
decision to publish, or preparation of the
manuscript.

**Competing interests:** The authors have declared
that no competing interests exist.

space medicine, since the start of space exploration projects up to nowadays. Although there exist some specialty journals, most of the literature in space medicine appears in other more general journals, including multidisciplinary ones. Given the peculiarity of the field, and limited possibilities of direct experimentation, much of the research comes from aviation and military studies [3].

Microgravity settings can be reproduced in free falling carriers, where the sum of all acting forces, other than gravity, is null or highly reduced. These environments for scientific and technological experimentation can be found both on Earth and in orbit. The principal carriers used in microgravity experimentation are drop towers and tubes, aircraft parabolic flights, sounding rockets, manned orbital platforms and automatic orbital platforms. Both conducting experiments in orbit or through weightless carriers on Earth is subject to important practical constraints, concerning the level and duration of the required microgravity settings and on the cost at which these experiments can be achieved [4].

Given these considerations, almost all the literature produced on the topic either involves very small samples or makes inferences from settings that could somehow resemble what happens in environments with zero or micro-gravity conditions (e.g. using clinostats, head down bed rests, saturation dives, etc.) [5]. Considering how difficult it is to collect data and results in space medicine, sharing information, experience and knowledge in a clear and transparent way would be of key importance for scientific progress in this field. One must take also into account that the attention towards this field is shifting more and more towards private corporate interests [6], so concerns might arise about the willingness of for-profit stakeholders to share new data for free.

Behind space exploration there is surely a strong scientific background, but it comes together with both political and corporate interests and considerable risk of bias given the intrinsic difficulties of carrying on rigorous scientific research on these particular topics [7]. In this regard, transparency features, such as sharing of data and code, availability of pre-registered protocols, as well as reporting of conflicts of interest and funding, are important to assess to evaluate the rigor of evidence obtained in biomedical research in this field [8–10]. Data and code sharing facilitate replication studies, allow checking for errors, and enable reuse of the data in secondary analyses and meta-analyses. Pre-registration allows comparing notes between protocols and published results. Reporting of conflicts of interest and funding also help to put the work in context and increase trust.

The aim of this work is to map the transparency of the space medicine literature by performing a meta-epidemiological assessment of transparency indicators in articles regarding scientific research in space medicine.

## Materials and methods

### Study sample

We examined all papers published in space medicine (as of January 25, 2023) in the PubMed Central Open Access (PMC OA) subset, since these are papers that can be massively downloaded for in depth text mining and manual analyses.

For the field of space medicine, we considered eligible all studies that aimed to study the effect of a non-terrestrial setting on terrestrial concepts. Studies that attempted to study any physiological or pathological changes in a space setting that could affect health of astronauts, people and living beings in general were also included. Any papers with medical or biological relevance in conditions that occur in space or simulating such environments were included. Only original research as well as reviews were eligible. Physics, chemistry and engineering and material science papers were excluded unless there was any medical or biological relevance

within their findings. Editorials, commentaries, letters to editors and any other kind of publication that can't be defined as an original article or a review were excluded.

The study protocol was pre-registered on Open Science Framework (OSF, https://doi.org/10.17605/OSF.IO/WN59C). No institutional ethics approval was requested since this is a literature-based review/evaluation.

## Search query

The search query applied in PubMed was the following: spaceflight* [ti] OR space flight* [ti] OR spacecraft* [ti] OR space craft* [ti] OR astronaut* [ti] OR microgravity [ti] OR space age [ti] OR nasa [ti] OR earth observation [ti] OR absence of gravity [ti] OR parabolic flight*[ti] OR space mission* [ti] OR space station* [ti] OR iss [ti] OR international space station* [ti] OR ESA [ti] OR Extraterrestrial [ti] OR Interplanetary [ti] OR Space crew* [ti] OR Space shuttle* [ti] OR Bioastronautic* [ti] OR Cosmonaut* [ti] OR Space medicine [ti] OR Planetary exploration* [ti] OR Space exploration* [ti].

After a screening of titles, abstracts, and, when needed, full texts, all original and review articles with medical or biological relevance in conditions that occur in space or that are simulating such environment were selected.

## Data extraction: Text mining

For each eligible article we used PubMed to extract information on metadata that includes PMID, PMCID, publication year, journal name, affiliation, and the R package rtransparent [9] to extract the following transparency indicators: (i) code sharing (ii) data sharing (iii) pre-registration, (iv) conflicts of interest and (v) funding statements. Searches used the rtransparent algorithms applied through the full text of the papers for specific words or phrases that strongly suggest that the aforementioned transparency indicators are present in that particular paper. The program uses regular expressions to adjust for variations in expressions.

## Manually extracted characteristics

From a random sample of 200 articles, selected through a random sampling method in R, we manually extracted additional characteristics: article type (original article, review, systematic review/meta-analysis); study setting (space shuttle or parabolic flights, simulated microgravity, other (astronaut, population screening (epidemiology studies), any other)); study design (interventional, non-interventional); type of data (simulated, real); study population (human (astronauts), human (general population), other (animal study, plants, in vitro, modeling-computer simulation, biological materials and components, genetics, hygiene, microbiology assessments, safety assessments (prevention or occupational hazards), any other)); funding institution (public institution, private institution non-profit, private institution for-profit organization); project initiator (academic institution, combination, private institution for profit organization, private institution non-profit, public institution, none, no funding statement); sample size; length of the article (number of pages); number of Tables, Figures, Appendices; and the statistical presentation of the abstract (reporting of P-values, reporting of confidence intervals or other uncertainty intervals). The categories for each extracted feature were prespecified in the protocol and after manual perusal of some articles we made some edits to the categories that would optimize data capture.

## Outcomes

Our primary outcomes were the 5 transparency indicators (data sharing, code sharing, pre-registration, funding, conflicts of interest) and the primary analysis was their change over time

using a binomial family generalized linear model. Secondary outcomes (also assessed for change over time) in the manually assessed sample focused on the project initiator, and funding institution, descriptively.

### Manual validation of transparency indicators

For the random sample of the 200 articles, the algorithmic outputs were validated manually. Subsequently the estimates of the proportion of articles satisfying each indicator were corrected for the rate of false positives and false negatives.

The corrected proportion C(i) of publications satisfying an indicator i was obtained by U(i) × TP + (1 − U(i)) × FN, where U(i) is the uncorrected proportion detected by the automated algorithm, TP is the proportion of true positives (proportion of those manually verified to satisfy the indicator among those identified by the algorithm as satisfying the indicator, and FN is the proportion of false negatives (proportion of those manually found to satisfy the indicator among those categorized by the algorithm not to satisfy the indicator).

For the random sample of papers that were found to contain a conflict of interest statement and funding disclosure the statements were assessed considering how many of them contained actual disclosures of specific conflicts or funding sources, respectively, and not merely a statement that there are no conflicts/funding, e.g. 'There is no conflict of interest', 'No funding was received' or 'Funding disclosure is not applicable'.

### Software

Text-mining was performed using rtransparent in R (R: The R Project for Statistical Computing. Available: https://www.r-project.org/). Analyses of time trends, 2xn tables and ANOVA [11] were performed in R. Statistical significance was claimed for p<0.005 with values 0.005–0.05 considered suggestive [12].

## Results

### Data extraction

A total of 2905 published items were retrieved in the PubMed search performed on January 25, 2023. Of these 883 were excluded, due to ineligibility in the primary screening, because they didn't discuss either space medicine or any biomedical related topic, or they were not original articles or reviews. 2022 records were left for further scrutiny. 798 were excluded for not being part of the PMC OA subset and 9 were excluded during the manual in-depth validation because they were realized to be commentaries, editorials, or perspective papers (Fig 1). A repository of all assessed papers evaluated in this paper—including the non-open access subset —is available as supplementary material (osf.io/89c72).

The median year of publication of the 1215 eligible articles was 2019 (interquartile range 2017 to 2021 and 1164 of them (95%) had been published in the last decade (2013–2022). The most frequent journals of publication of these 1215 eligible articles were NPJ Microgravity, Scientific Reports, PLoS One, International Journal of Molecular Sciences, and Frontiers in Physiology.

### Transparency indicators

In the 1215 included articles, 39 (3%) shared code, 258 (21%) shared data, 10 (1%) were registered, 110 (90%) contained a conflict-of-interest statement, and 1141 (93%) included a funding statement.

On manual validation, 191 articles were evaluated (9 were excluded as they were found not to be original or review articles upon in-depth assessment). Among the 191 articles, the false-

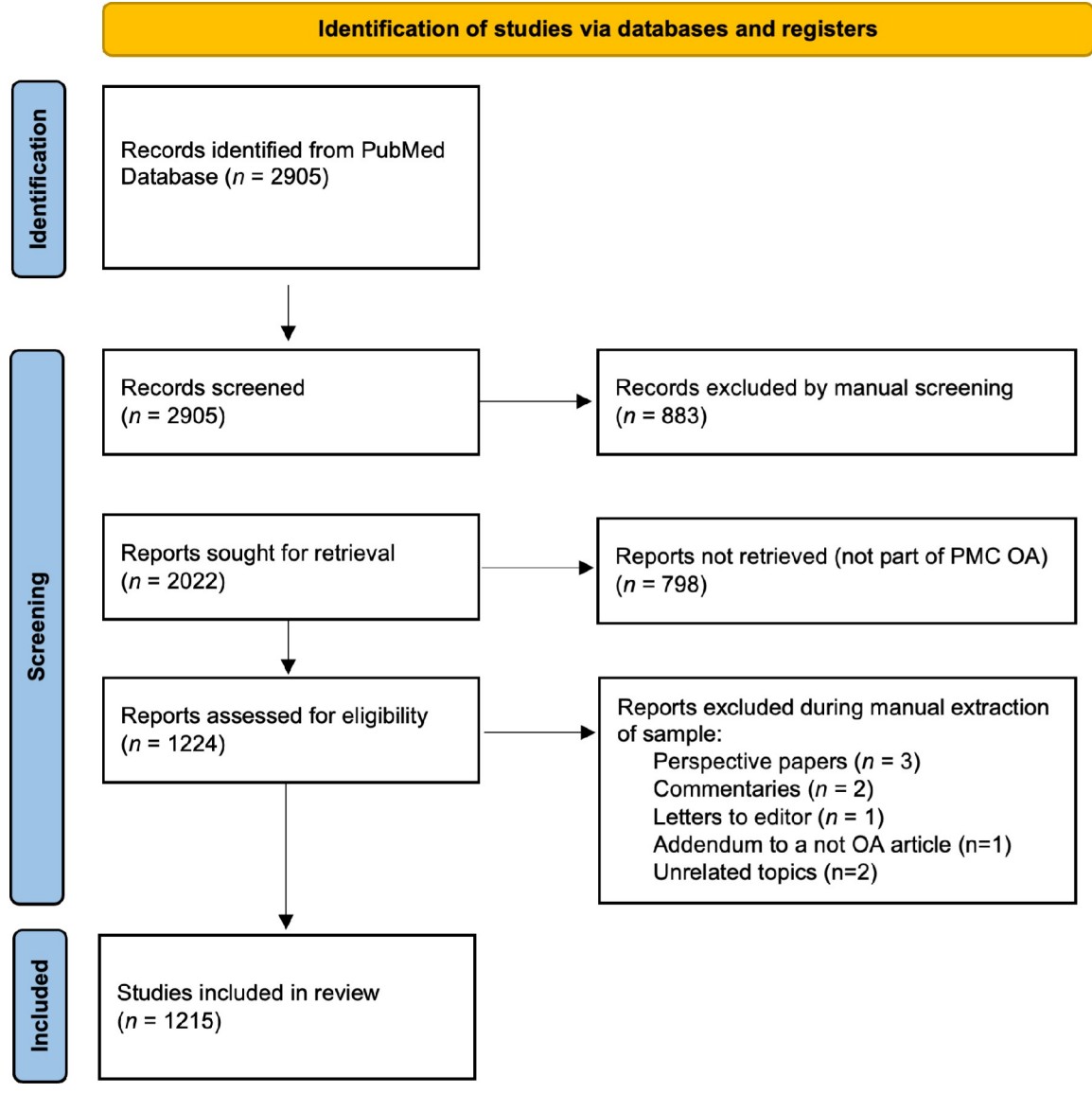

**Fig 1. Study selection flowchart.**

positive rate of the algorithm was 1/7 (14%) for code sharing, 1/38 (3%) for data sharing, and 0/1 (0%) for registration. The respective false-negative rates were 4/184 (2%), 13/153 (8%), and 0/190 (0%). Adjusting for the manual validation results, the corrected proportions of transparency indicators were 5% for code sharing, 27% for data sharing, and 1% for registrations.

Eight of 178 (4%) of funding disclosures practically stated that there was no funding for the study. One hundred sixty-eight of 184 (91%) of the conflict-of-interest statements practically declared that there was no conflict of interest.

## Time trend of transparency

Time trends of transparency were primarily evaluated in the last decade (2013–2022) when there were more than 20 eligible papers each year (in earlier years very few articles were eligible in the PMC OA subset). The choice of the time period was made to observe trends and ensure

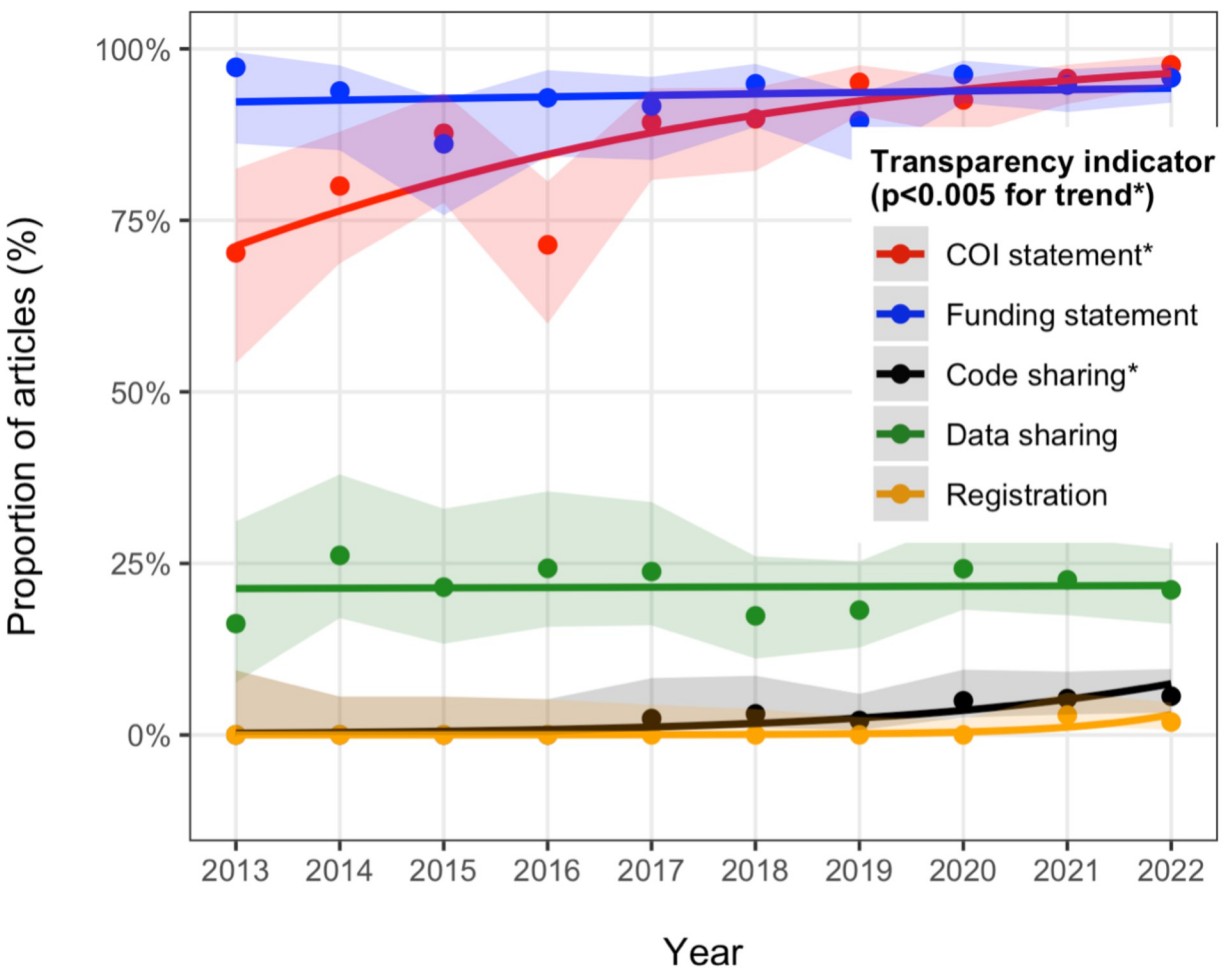

**Fig 2. Evolution of transparency indicators over time.** Only the last decade (2013–2022) is shown, since there were few eligible articles before 2013 and confidence intervals for these earlier years are very wide. * Denotes a statistically significant time trend at p<0.005.

a sufficient number of publications for meaningful analysis. Before 2013, the number of relevant papers was too low to infer any trends in transparency indicators with any certainty. The primary comparisons of transparency indicators between 2013 and 2022 showed a statistically significant increase in code sharing (p = 0.0003), conflict of interest statements (p<0.00001), and potentially also registration (p = 0.01), but not data sharing (p = 0.95) or funding disclosures (p = 0.1). Consideration of all eligible articles for all publication years yielded similar results (p = 0.0003 for code sharing, p<0.00001 for conflicts of interest, 0.0003 for funding disclosures, 0.01 for registration, and 0.15 for data sharing) (Fig 2).

The improvements during the last decade were modest in absolute magnitude for code sharing and funding disclosures (both ≤6%), while for conflict-of-interest statements the absolute percentage change was 28% in the fit regression lines.

## Manually extracted characteristics

In the random sample of 200 articles (191 eligible) that were examined in depth, most were classified as original research (n = 156, 82%), while the other were reviews (n = 35, (18%), with only one systematic review and the rest being narrative reviews). Reviews had no sharing of

code or data or registration, with the exception of one review sharing data. Also among reviews, 31% did not have funding disclosures. Excluding reviews, original articles shared data 32% of the time and almost always had conflicts of interest and funding disclosures.

In our sample, space flight or parabolic flight (40%) and simulated microgravity (38%) accounted for most papers. There was more data sharing (46%) in the space/parabolic literature (p<0.0001 by chi-square test).

Two thirds of the studies were interventional and all studies without funding disclosures were non-interventional (p<0.0001 by Fisher's exact test).

Most studies were not on humans (74%), while the human studies involved astronauts in 13% of the cases and 14% were conducted on non-astronaut populations. There were no notable differences in transparency indicators based on type of participants.

Almost all studies presented no $P$ values or confidence intervals (CIs) in their abstract (94% and 98% respectively). Most studies were publicly funded and the project initiator was almost always a university or government institution. Private entities were rarely involved in these published papers. None of these features were statistically significantly associated with any of the transparency indicators. There was no statistically significant change over time in the proportion of trials with public funding and in the proportion with a university initiator.

The full extracted characteristics of the categorical extracted variables can be seen in Table 1 and the raw data is found in S1 Table in osf.io/89c72.

Among the continuous characteristics assessed manually in the randomly selected papers (Table 2), the median (IQR) sample size was 14 (IQR 9 to 27), and the selected papers were mostly lengthy articles. Exploratory ANOVA showed more appendices/supplementary files among papers that shared their data (p<0.0001), and lengthier articles among those with a conflict of interest statement (p<0.0001); the other probed associations were not statistically significant.

**Most transparent papers.**   Our approach enabled us to identify the most transparent papers (n = 26) across our sample. All of them provided open data and open code. Funding and conflict of interest were also coherently made explicit across all these papers. Nevertheless, none of them were preregistered. These papers consist of original articles on research conducted in space, with fifteen focusing on microbiological or genomic assessments of samples collected on the International Space Station or other spacecraft, and the others addressing various basic science experiments.

## Discussion

This evaluation of 1,215 open-access papers in the field of space medicine revealed that less than a third shared their data, code sharing was rare, and registration almost non-existent. Almost all articles included conflicts of interest and funding disclosures, but very few mentioned specific conflicts. There was significant improvement over time in several of these transparency indicators, but not for data sharing. Space/parabolic flight papers had the highest rate of data sharing, but, even among these papers, more than half did not share their data. The examined literature lies almost entirely in the space of public, not-for-profit funders and with the initiators being academic or governmental institutions with a dearth of papers coming from the corporate sector.

In 2014, in response to the Executive Office of the President's February 22, 2013, Office of Science and Technology Policy (OSTP) Memorandum for the Heads of Executive Departments and Agencies, "Increasing Access to the Results of Federally Funded Scientific Research.", NASA issued a plan for increasing access to results of scientific research (https://ntrs.nasa.gov/citations/20150020926). This strategy broadened NASA's commitment to open

**Table 1. Categorical characteristics of studies evaluated manually and their relationship with transparency indicators.**

| Correlates | N/Total (%) | Code sharing | Data sharing | Registration | Conflict of interest statement | Funding disclosure |
|---|---|---|---|---|---|---|
| *Study type* | | | | | | |
| Original article | 155/191(81) | 10/155 (6) | 49/155 (32) | 1/155 (1) | 146/155 (94) | 153/155 (99) |
| Review | 35/191(18) | 0/35 (0) | 1/35 (3) | 0/35 (0) | 32/35 (91) | 24/35 (69) |
| Systematic Review/Meta-Analysis | 1/191(1) | 0/1 (0) | 0/1 (0) | 0/1 (0) | 1/1 (100) | 1/1 (100) |
| *Study setting* | | | | | | |
| Space flight/Parabolic flight | 76/191(40) | 4/76 (5) | 35/76 (46) | 0/76 (0) | 71/76 (93) | 76/76 (100) |
| Simulated microgravity | 73/191(38) | 4/73 (5) | 10/73 (14) | 1/73 (1) | 71/73 (97) | 71/73 (97) |
| Other | 42/191(22) | 2/42 (5) | 5/42 (12) | 0/42 (0) | 37/42 (88) | 31/42 (74) |
| *Interventional* | | | | | | |
| Interventional | 127/191(66) | 3/127 (2) | 34/127 (27) | 1/127 (1) | 123/127 (97) | 127/127 (100) |
| Non interventional | 64/191(34) | 7/64 (11) | 16/64 (25) | 0/64 (0) | 56/64 (88) | 51/64 (80) |
| *Study population* | | | | | | |
| Human (Astronauts) | 24/191(13) | 1/24 (4) | 6/24 (25) | 0/24 (0) | 22/24 (92) | 22/24 (92) |
| Human (general population) | 26/191(14) | 1/26 (4) | 4/26 (15) | 1/26 (4) | 24/26 (92) | 25/26 (96) |
| Other | 141/191(74) | 8/141 (6) | 40/141 (28) | 0/141 (0) | 133/141 (94) | 131/141 (93) |
| *P-value in abstract* | | | | | | |
| Yes | 12/191(6) | 0/12 (0) | 3/12 (25) | 1/12 (8) | 11/12 (92) | 12/12 (100) |
| No | 179/191(94) | 10/179 (6) | 47/179 (26) | 0/179 (0) | 168/179 (94) | 166/179 (93) |
| *Uncertainty interval in abstract* | | | | | | |
| Yes | 4/191(2) | 0/4 (0) | 1/4 (25) | 0/4 (0) | 4/4 (100) | 4/4 (100) |
| No | 187/191(98) | 10/187 (5) | 49/187 (26) | 1/187 (1) | 175/187 (94) | 174/187 (93) |
| *Funder* | | | | | | |
| Academic institution | 5/191(3) | 1/5 (20) | 1/5 (20) | 0/5 (0) | 5/5 (100) | 5/5 (100) |
| Combination* | 36/191(19) | 3/36 (8) | 12/36 (33) | 0/36 (0) | 34/36 (94) | 36/36 (100) |
| Private institution for profit organization | 1/191(1) | 0/1 (0) | 1/1 (100) | 0/1 (0) | 1/1 (100) | 1/1 (100) |
| Private institution non-profit | 1/191(1) | 0/1 (0) | 0/1 (0) | 0/1 (0) | 1/1 (100) | 1/1 (100) |
| Public institution | 129/191(68) | 5/129 (4) | 36/129 (28) | 1/129 (1) | 120/129 (93) | 127/129 (98) |
| None | 6/191(3) | 0/6 (0) | 0/6 (0) | 0/6 (0) | 6/6 (100) | 6/6 (100) |
| No funding statement | 13/191(5) | 1/13 (8) | 0/13 (0) | 0/13 (0) | 12/13 (92) | 2/13 (15) |
| *Data type* | | | | | | |
| Real | 152/191(80) | 7/152 (5) | 47/152 (31) | 1/152 (1) | 143/152 (94) | 151/152 (99) |
| Simulated | 5/191(3) | 3/5 (60) | 2/5 (40) | 0/5 (0) | 5/5 (100) | 4/5 (80) |
| Narrative only | 34/191(18) | 0/34 (0) | 1/34 (3) | 0/34 (0) | 31/34 (91) | 23/34 (68) |
| *Project initiator* | | | | | | |
| Government institution | 56/191(29) | 6/56 (11) | 17/56 (30) | 1/56 (2) | 53/56 (95) | 54/56 (96) |
| Hospital | 1/191(1) | 0/1 (0) | 1/1 (100) | 0/1 (0) | 1/1 (100) | 1/1 (100) |
| Private company | 6/191(3) | 0/6 (0) | 4/6 (67) | 0/6 (0) | 5/6 (83) | 5/6 (83) |
| University | 128/191(67) | 4/128 (3) | 28/128 (22) | 0/128 (0) | 120/128 (94) | 118/128 (92) |

*Only one paper has a combination of private for-profit + government institution. The majority have a combination government (NASA, ESA, etc.) + academic, except for two papers that have a combination of government + non-profit.

access, encompassing both data and publications related to all scientific research supported by the Agency. NASA also encourages public access to scientific publications through the Public Access initiative, which is part of the agency's framework for increasing public access to scientific publications and digital scientific data (https://sti.nasa.gov/research-access/).

The European Space Agency has been addressing the issue too, by establishing the electronic support infrastructure for gathering, storing, disseminating, and processing healthcare

**Table 2. Continuous characteristics of studies evaluated manually and their relationship with transparency indicators.**

|  | Overall | Code sharing | Data sharing | Registered | Conflict of interest statement | Funding disclosure |
|---|---|---|---|---|---|---|
| Sample size | 14 (9–27) | 89 (89–89) | 16 (12–32) | 20 (20–20) | 15 (10–28) | 14 (9–24.75) |
| Article length | 12 (10–17) | 14 (12.25–19) | 14 (11–17) | 17 (17–17) | 13 (10–17) | 13 (10–17) |
| Number of tables | 1 (0–3) | 1 (0–1) | 1 (0.75–3) | 5 (5–5) | 1 (0–3) | 1 (0–3) |
| Number of figures | 5 (3–7) | 6 (4.25–7.75) | 5 (4–6.75) | 4 (4–4) | 5 (3–7) | 5 (4–7) |
| Number of appendices/supplementary files | 0 (0–1) | 2 (1–4) | 2 (1–4) | 1 (1–1) | 1 (0–1) | 1 (0–1) |

All presented as Median (IQR)

data within the contemporary realm of eHealth. Comparable solutions have been developed to address analogous issues in the management of medical data, both in terrestrial and space environments (https://www.esa.int/Enabling_Support/Preparing_for_the_Future/Space_for_Earth/Space_for_health/Medical_data_management).

he selection of the 5 assessed indicators was based on their broad applicability and recognized effectiveness in improving transparency across various fields. While they may not be equally relevant to all studies in space life science, they address key aspects that enhance research quality and credibility. Furthermore, we based the selection of those indicators following previous studies performed using the same R package, in order to be able to compare the results across different fields. This also allows to draw conclusions by comparing results and assess how transparency changes across different fields.

The importance of transparency and public access to research in science has raised interest and awareness among scientists from very different backgrounds during the past years. Serghiou et al. recently [9] presented an open-source, automated approach to identify five indicators of transparency, which was also utilized in this study. The evaluation of 2,751,420 open-access PMC articles (from 1959 to 2020) suggested substantial improvements in reporting conflicts of interest (COI) and funding disclosures over time, across various scientific fields and publication venues. However, data sharing, code sharing, and protocol registration still lag significantly. The rates of code and data sharing that we observed for space medicine are in the same range to what has been observed in the evaluation of the overall scientific literature in biomedicine [13].

When compared to transparency indicators in infectious disease models, another medical field where in-depth assessments have been conducted in the past [14], our sample showed lower rates of code sharing (3% versus 21.5%), but similar rates of data sharing. Another paper[15] assessed transparency of COVID-19-related research in dental journals. Out of 650 articles published across 59 dental journals, 74% included disclosure of conflicts of interest, 40% revealed funding sources, and 4% had preregistration. Only one study, representing 0.15%, shared raw data, and none provided access to their code. Articles published in journals with higher impact factors, as well as those published in 2020, demonstrated a greater tendency towards transparency compared to those in 2021 [15]. Another transparency assessment was performed in orthopedic literature [16]. Out of 286 publications, 13.3% lacked a conflict-of-interest statement and about half (49.0%) did not mention funding. Of 182 empirical studies, 95.6% were not preregistered, and only 8 provided preregistration statements. 8.8% of these studies offered a data availability statement, with 13 datasets accessible, but only 2 providing complete raw data for reproducing results. None included an analysis script, and only 1.1% granted access to their study protocol. When compared against other specialty journals, space medicine research shows overall a better performance in providing conflicts of interest and funding disclosures, while data sharing seems suboptimal and protocol registration is rare across very diverse biomedical fields.

Regarding data sharing, one might expect lower rates in some types of space experiments due to legal and privacy issues given that astronauts are a well-defined group. Surprisingly, experiments on astronauts did not show lower rates of data sharing compared to non-astronaut experiments (they had a non-significant trend of higher sharing). Moreover, we documented a substantially higher rate of data sharing rate in the space subset sample compared to the microgravity subset, which may be attributed to the aforementioned NASA Plan for Increasing Access to the Results of Scientific Research and ESA's commitment to improve research open access. We recorded very low rates of code sharing, however it should be acknowledged that both in space medicine and in other fields, a large share of articles does not use any code anyhow, while a few papers may have proprietary code that cannot be shared.

The lowest transparency was observed within the sampled review articles, with only one review conducted systematically and practically no attention given by reviews to data or code sharing or protocol registration. This is worrisome because reviews have an important mission of integrating information and guiding future work. Systematic reviews have long been established as the norm across biomedicine and it is routinely recommended that they should be registered [17]. Access to their data is important to allow verifying and reusing this information. Unfortunately, many reviews, including systematic reviews, across biomedicine, are of very low quality and misleading [18].

Lack of protocol registration, particularly for space-based experiments, is concerning. These experiments often involve small samples and require significant time and financial resources. To avoid resource wastage, it is crucial for such research to follow a rigorous and efficient methodology that includes drafting and registering a protocol before conducting these experiments [19, 20]. Registration remains notoriously uncommon in space medicine, as is in most scientific fields, except for those that involve randomized controlled trials [13, 21] and systematic reviews [17]. However, there have been initiatives and suggestions on improving additional types of studies more frequently, including but not limited to, observational studies [22–27], animal studies [20], and mathematical modeling [10, 19, 28].

Several limitations need to be considered in the interpretation of the findings from this study. First, the restriction to open access papers in our analysis may introduce a potential source of bias. Nevertheless, it is unlikely that open access papers perform worse on transparency indicators than those that are not open access. Open access is in fact another dimension of transparency [9]. Second, our search aimed to have high specificity and thus retrieve articles that had a high likelihood of being relevant to space medicine. However, it did not capture all papers that pertain to space medicine. We used a search that focused on papers that have pertinent terms in their title, but many other papers may be relevant to space medicine without having these terms explicitly in their titles. Moreover, we focused on PubMed OA papers that could be readily text-mined, but other approaches for obtaining open access papers, e.g. OpenAlex, might have yielded some additional papers. There is no strong reason to believe, nevertheless, that these additional papers would have better transparency features. Furthermore, despite low error rates, the rtransparent algorithm may not capture all variations in reporting and its nuances. The algorithm also presumes that a reported link is accessible, and the code reproducible which may not necessarily be the case. Moreover, the manually assessed articles, although representative, may not fully capture the diversity of space medicine research, and the findings may not be generalizable to the entire field. In particular, we were impressed how little of the examined publication corpus reflected work initiated and/or supported by for-profit corporations. These corporations currently invest large amounts of funding to space-related research. However, apparently their work is not typically released in peer-reviewed journal venues. This means that our estimates about transparency in the field may be overestimates if unpublished work is also taken into perspective. "Stealth research", where

corporations perform research without communicating it in the peer-reviewed literature, may allow even fraud to be introduced and be difficult to detect, as has been shown in other fields of entrepreneurship and start-ups like Theranos [29, 30].

In conclusion, space medicine is a unique and evolving field with a complex interplay of scientific, political, and corporate interests. To advance scientific progress in this field, transparent sharing of information, experiences, and knowledge is of paramount importance, particularly considering the changing landscape of stakeholders, including private corporations. Our study shows that transparency overall is suboptimal, albeit with some positive time trends. As space exploration continues to evolve, fostering a culture of transparency and collaboration among stakeholders is essential to ensure the integrity and advancement of space medicine research in the years to come. Multiple stakeholders, including funding agencies, journals, scientific societies and grassroot movements from scientists may help make research more transparent. These stakeholders may coordinate efforts and reach greater consensus on best research practices, including what items and transparency indicators should be mandatory for publication.

## Author Contributions

**Conceptualization:** Rosa Katia Bellomo, John P. A. Ioannidis.

**Data curation:** Rosa Katia Bellomo.

**Formal analysis:** Emmanuel A. Zavalis.

**Investigation:** Rosa Katia Bellomo, Emmanuel A. Zavalis.

**Methodology:** John P. A. Ioannidis.

**Project administration:** John P. A. Ioannidis.

**Supervision:** John P. A. Ioannidis.

**Writing – original draft:** Rosa Katia Bellomo, Emmanuel A. Zavalis.

**Writing – review & editing:** John P. A. Ioannidis.

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
