## [Decision Letter · Decision Letter 0]

5 Feb 2024

PONE-D-24-00779Assessment of transparency indicators in Space MedicinePLOS ONE

Dear Dr. Bellomo,

Thank you for submitting your manuscript to PLOS ONE. After careful consideration, we feel that it has merit but does not fully meet PLOS ONE’s publication criteria as it currently stands. Therefore, we invite you to submit a revised version of the manuscript that addresses the points raised during the review process.

We look forward to receiving your revised manuscript.

Kind regards,

Tadej Debevec, Ph.D.

Academic Editor

PLOS ONE

Journal Requirements:

"The work of John Ioannidis is supported by an unrestricted gift from Sue and Bob O’Donnell to Stanford. The work of Rosa Katia Bellomo is funded within the framework of the project “ExACT - European network staff eXchange for integrating precision health in the health Care sysTems” (Grant Agreement n. 823995) funded by the European Commission under the H2020 – Marie Slodowska Curie Action – RISE scheme."

Please state what role the funders took in the study.  If the funders had no role, please state: ""The funders had no role in study design, data collection and analysis, decision to publish, or preparation of the manuscript."" If this statement is not correct you must amend it as needed. 

3. Please be informed that funding information should not appear in the Acknowledgments section or other areas of your manuscript. We will only publish funding information present in the Funding Statement section of the online submission form. Please remove any funding-related text from the manuscript. 

4. In this instance it seems there may be acceptable restrictions in place that prevent the public sharing of your minimal data. However, in line with our goal of ensuring long-term data availability to all interested researchers, PLOS’ Data Policy states that authors cannot be the sole named individuals responsible for ensuring data access (http://journals.plos.org/plosone/s/data-availability#loc-acceptable-data-sharing-methods).

**Additional Editor Comments:**

As you will see below the manuscript has been reviewed by two referees and while ref#1 was rather scant, the points raised by reviewer #2 should be carefully and fully addressed in the revision to ensure the paper fully aligns with PLOS ONE’s publication

Reviewers' comments:

Reviewer's Responses to Questions

**Comments to the Author**

1. Is the manuscript technically sound, and do the data support the conclusions?

Reviewer #1: Partly

Reviewer #2: Yes

2. Has the statistical analysis been performed appropriately and rigorously? 

Reviewer #1: Yes

Reviewer #2: Yes

3. Have the authors made all data underlying the findings in their manuscript fully available?

Reviewer #1: Yes

Reviewer #2: Yes

4. Is the manuscript presented in an intelligible fashion and written in standard English?

Reviewer #1: Yes

Reviewer #2: Yes

5. Review Comments to the Author

Reviewer #1: There few comments to make because the paper is well preformed, described and discussed, however I would suggest to change the source of the papers, I am using OpenAlex, and the number of papers is much more bigger using similar seach query than the ones found by the use of PubMed OA, this could be an important bias for the conclusions.

I would also liked to state specific aims of the work,

Reviewer #2: The manuscript presented by Bellomo et al., is an interesting topic that certainly needs to be discussed in order to maintain scientific rigor. The writing (grammar and style) in general is good, a final proof read will pick up a couple of typos. I would like to thank the authors for an enjoyable read and ask that they consider several minor comments. Further I command them on their extensive work.

While the title refers to Space Medicine, I found that the ms uses indicators of transparency that are maybe more applicable to other fields. By that I mean that in space medicine or space life science not all studies conducted require code and thus code sharing in this field will be reduced.

How many papers in space medicine or space life sciences actually use code? Were any papers included where the code was noted as propriety and therefore could not be shared?

Was there a reason that bed rest, head down tilt or dry immersion were not included in the search query? However, I do see from the chosen articles in supplementary material that some bed rest papers were included.

How did the authors choose the time period (2013-2023) in which to select papers from? Including this justification in the paper would aid the reader.

In terms of observing trends (Fig. 2) a longer time period would be of interest. Although I fully understand that increasing the time span is a significant undertaking that may not change the outcome of the paper.

Many journals actually require conflict of interest and funding statements but obviously do not require the same for data /code sharing which very likely accounts for the higher % of COI and funding statements rather than authors trying to hide something.

Can the authors add some information on why they feel these particular markers improve transparency?

What is the end goal of data sharing? To double check the data or to start to encourage study replication?

Can the authors provide any guidelines or recommendations on how to be more transparent? Does this stem from the journals or from the scientists?

What kind of items/transparency indicators do the authors recommend become a mandatory requirement for publication?

How do the authors interpret the meaningfulness of the selected 5 indicators in improving transparency in space life science research? Are these the most important or useful?

It would be of interest to include how the subset of 200 articles chosen?

6. PLOS authors have the option to publish the peer review history of their article (what does this mean?). If published, this will include your full peer review and any attached files.

Reviewer #1: **Yes: **Remedios Melero

Reviewer #2: No

---

## [Author Response · Author response to Decision Letter 0]

13 Feb 2024

The manuscript was formatted as required.

"The work of John Ioannidis is supported by an unrestricted gift from Sue and Bob O’Donnell to Stanford. The work of Rosa Katia Bellomo is funded within the framework of the project “ExACT - European network staff eXchange for integrating precision health in the health Care sysTems” (Grant Agreement n. 823995) funded by the European Commission under the H2020 – Marie Slodowska Curie Action – RISE scheme."

Please state what role the funders took in the study. If the funders had no role, please state: ""The funders had no role in study design, data collection and analysis, decision to publish, or preparation of the manuscript."" If this statement is not correct you must amend it as needed. 

We updated the cover letter according to your requirements.

3. Please be informed that funding information should not appear in the Acknowledgments section or other areas of your manuscript. We will only publish funding information present in the Funding Statement section of the online submission form. Please remove any funding-related text from the manuscript. 

All funding related text was removed from the manuscript.

4. In this instance it seems there may be acceptable restrictions in place that prevent the public sharing of your minimal data. However, in line with our goal of ensuring long-term data availability to all interested researchers, PLOS’ Data Policy states that authors cannot be the sole named individuals responsible for ensuring data access (http://journals.plos.org/plosone/s/data-availability#loc-acceptable-data-sharing-methods).

The analyzed information dataset is available at OSF (www.doi.org/10.17605/OSF.IO/YXU9Q.

Additional Editor Comments:

As you will see below the manuscript has been reviewed by two referees and while ref#1 was rather scant, the points raised by reviewer #2 should be carefully and fully addressed in the revision to ensure the paper fully aligns with PLOS ONE’s publication

Comments to the Author

1. Is the manuscript technically sound, and do the data support the conclusions?

Reviewer #1: Partly

Reviewer #2: Yes

2. Has the statistical analysis been performed appropriately and rigorously?

Reviewer #1: Yes

Reviewer #2: Yes

3. Have the authors made all data underlying the findings in their manuscript fully available?

Reviewer #1: Yes

Reviewer #2: Yes

4. Is the manuscript presented in an intelligible fashion and written in standard English?

Reviewer #1: Yes

Reviewer #2: Yes

5. Review Comments to the Author

Reviewer #1: There few comments to make because the paper is well preformed, described and discussed, however I would suggest to change the source of the papers, I am using OpenAlex, and the number of papers is much more bigger using similar seach query than the ones found by the use of PubMed OA, this could be an important bias for the conclusions.

We recognize that different tools may yield different numbers of articles. We have added in the limitations section in the Discussion after the statement “Open access is in fact another dimension of transparency(9).” the following new text: “Second, our search aimed to have high specificity and thus retrieve articles that had a high likelihood of being relevant to space medicine. However, it did not capture all papers that pertain to space medicine. We used a search that focused on papers that have pertinent terms in their title, but many other papers may be relevant to space medicine without having these terms explicitly in their titles. Moreover, we focused on PubMed OA papers that could be readily text-mined, but other approaches for obtaining open access papers, e.g. OpenAlex, might have yielded some additional papers. There is no strong reason to believe, nevertheless, that these additional papers would have better transparency features.” 

I would also liked to state specific aims of the work,

We have replaced the last sentence of the Introduction (previously: “In order to map the transparency of the space medicine literature, here we performed a meta-epidemiological assessment of these transparency indicators in articles regarding scientific research in space medicine.”) with the more explicit: “The aim of this work is to map the transparency of the space medicine literature by performing a meta-epidemiological assessment of transparency indicators in articles regarding scientific research in space medicine.”

Reviewer #2: The manuscript presented by Bellomo et al., is an interesting topic that certainly needs to be discussed in order to maintain scientific rigor. The writing (grammar and style) in general is good, a final proof read will pick up a couple of typos. I would like to thank the authors for an enjoyable read and ask that they consider several minor comments. Further I command them on their extensive work.

1. While the title refers to Space Medicine, I found that the ms uses indicators of transparency that are maybe more applicable to other fields. By that I mean that in space medicine or space life science not all studies conducted require code and thus code sharing in this field will be reduced.

How many papers in space medicine or space life sciences actually use code? Were any papers included where the code was noted as propriety and therefore could not be shared?

We have added at the end of the paragraph in the Discussion that starts with “Regarding data sharing…” the following: “We recorded very low rates of code sharing, however it should be acknowledged that both in space medicine and in other fields, a large share of articles does not use any code anyhow, while a few papers may have proprietary code that cannot be shared.” 

2. Was there a reason that bed rest, head down tilt or dry immersion were not included in the search query? However, I do see from the chosen articles in supplementary material that some bed rest papers were included.

Thank you for your insightful question. Our paper focuses on space medicine and microgravity, encompassing both real space conditions and simulations achieved on Earth through experiments like bed rest, head down tilt, and dry immersion. While we didn't specifically include the terms "bed rest," "head down tilt," or "dry immersion" in our search query to avoid potential noise in the results, our search criteria were designed to capture all articles with medical or biological relevance in conditions that occur in space or simulate such environments. As a result, articles describing terrestrial experiments simulating microgravity, including those you mentioned, were indeed included in our research and some of them related to concepts such as bed rest. Please see also our reply to the first comment of reviewer 1 that explains that we used search terms with high specificity. 

3. How did the authors choose the time period (2013-2023) in which to select papers from? Including this justification in the paper would aid the reader.

In terms of observing trends (Fig. 2) a longer time period would be of interest. Although I fully understand that increasing the time span is a significant undertaking that may not change the outcome of the paper.

Many journals actually require conflict of interest and funding statements but obviously do not require the same for data /code sharing which very likely accounts for the higher % of COI and funding statements rather than authors trying to hide something.

The choice of the time period (2013-2023) was made to observe trends and ensure a sufficient number of publications for meaningful analysis. Before 2013, the number of relevant papers was too low to infer any trends in transparency indicators with any certainty. The adoption of platforms like Github in the research community also began around that time. The revised manuscript now states:

“The choice of the time period was made to observe trends and ensure a sufficient number of publications for meaningful analysis. Before 2013, the number of relevant papers was too low to infer any trends in transparency indicators with any certainty.”

4. Can the authors add some information on why they feel these particular markers improve transparency?

In the Introduction, following “In this regard, transparency features, such as sharing of data and code, availability of pre-registered protocols, as well as reporting of conflicts of interest and funding, are important to assess to evaluate the rigor of evidence obtained in biomedical research in this field(8–10)”, we have added: “Data and code sharing facilitate replication studies, allow checking for errors, and enable reuse of the data in secondary analyses and meta-analyses. Pre-registration allows comparing notes between protocols and published results. Reporting of conflicts of interest and funding also help to put the work in context and increase trust.”

What is the end goal of data sharing? To double check the data or to start to encourage study replication?

As above, we have clarified that “Data and code sharing facilitate replication studies, allow checking for errors, and enable reuse of the data in secondary analyses and meta-analyses.”

5. Can the authors provide any guidelines or recommendations on how to be more transparent? Does this stem from the journals or from the scientists?

What kind of items/transparency indicators do the authors recommend become a mandatory requirement for publication?

Transparency and reproducibility are becoming more and more important in scientific research as article production is becoming incredibly extensive and prolific. We have added in the Discussion at the end of the Conclusions that “Multiple stakeholders, including funding agencies, journals, scientific societies and grassroot movements from scientists may help make research more transparent. These stakeholders may coordinate efforts and reach greater consensus on best research practices, including what items and transparency indicators should be mandatory for publication.” We would prefer to avoid making recommendations and guidelines just by ourselves. 

6. How do the authors interpret the meaningfulness of the selected 5 indicators in improving transparency in space life science research? Are these the most important or useful?

We have added in the Discussion a new paragraph, which is now the 4th paragraph in this section: “The selection of the 5 assessed indicators was based on their broad applicability and recognized effectiveness in improving transparency across various fields. While they may not be equally relevant to all studies in space life science, they address key aspects that enhance research quality and credibility. Furthermore, we based the selection of those indicators following previous studies performed using the same R package, in order to be able to compare the results across different fields. This also allows to draw conclusions by comparing results and assess how transparency changes across different fields.” 

7. It would be of interest to include how the subset of 200 articles chosen?

The subset of 200 articles was selected through a random sampling method in R. Manuscript integrated accordingly at line 169 “selected through a random sampling method in R,”.

6. PLOS authors have the option to publish the peer review history of their article (what does this mean?). If published, this will include your full peer review and any attached files.

Do you want your identity to be public for this peer review? For information about this choice, including consent withdrawal, please see our Privacy Policy.

Reviewer #1: Yes: Remedios Melero

Reviewer #2: No

---

## [Decision Letter · Decision Letter 1]

5 Mar 2024

Assessment of transparency indicators in Space Medicine

PONE-D-24-00779R1

Dear Dr. Bellomo,

We’re pleased to inform you that your manuscript has been judged scientifically suitable for publication and will be formally accepted for publication once it meets all outstanding technical requirements.

Kind regards,

Tadej Debevec, Ph.D.

Academic Editor

PLOS ONE

Additional Editor Comments (optional):

Reviewers' comments:

Reviewer's Responses to Questions

**Comments to the Author**

1. If the authors have adequately addressed your comments raised in a previous round of review and you feel that this manuscript is now acceptable for publication, you may indicate that here to bypass the “Comments to the Author” section, enter your conflict of interest statement in the “Confidential to Editor” section, and submit your "Accept" recommendation.

Reviewer #1: All comments have been addressed

Reviewer #2: All comments have been addressed

2. Is the manuscript technically sound, and do the data support the conclusions?

Reviewer #1: Yes

Reviewer #2: Yes

3. Has the statistical analysis been performed appropriately and rigorously? 

Reviewer #1: Yes

Reviewer #2: Yes

4. Have the authors made all data underlying the findings in their manuscript fully available?

Reviewer #1: Yes

Reviewer #2: Yes

5. Is the manuscript presented in an intelligible fashion and written in standard English?

Reviewer #1: Yes

Reviewer #2: Yes

6. Review Comments to the Author

Reviewer #1: I think all issues/questions raised by reviewers have been accomplished, and responded by the authors.

Reviewer #2: I would like to thank you for addressing my comments,

I wish you all the best with your future work

7. PLOS authors have the option to publish the peer review history of their article (what does this mean?). If published, this will include your full peer review and any attached files.

Reviewer #1: **Yes: **Remedios Melero

Reviewer #2: No

---

## [Editor Report · Acceptance letter]

23 Mar 2024

PONE-D-24-00779R1 

PLOS ONE

Dear Dr. Bellomo, 

I'm pleased to inform you that your manuscript has been deemed suitable for publication in PLOS ONE. Congratulations! Your manuscript is now being handed over to our production team.

Kind regards, 

on behalf of

Dr. Tadej Debevec 

Academic Editor

PLOS ONE